# Where does In-context Learning Happen in Large Language Models?

**Suzanna Sia** *
Johns Hopkins University
ssia1@jh.edu

**David Mueller**
Johns Hopkins University
dam@cs.jhu.edu

**Kevin Duh**
Johns Hopkins University
kevinduh@cs.jhu.edu

## Abstract

Self-supervised large language models have demonstrated the ability to perform various tasks via in-context learning, but little is known about where the model locates the task with respect to prompt instructions and demonstration examples. In this work, we attempt to characterize the region where large language models transition from recognizing the task to performing the task. Through a series of layer-wise context-masking experiments on GPTNEO2.7B, BLOOM3B, and STARCODER2-7B, LLAMA3.1-8B, LLAMA3.1-8B-INSTRUCT, on Machine Translation and Code generation, we demonstrate evidence of a "task recognition" point where the task is encoded into the input representations and attention to context is no longer necessary. Taking advantage of this redundancy results in 45% computational savings when prompting with 5 examples, and task recognition achieved at layer 14 / 32 using an example with Machine Translation. Our findings also have implication for resource and parameter efficient fine-tuning; we observe a correspondence between fine-tuning performance of individual LoRA layers and the task recognition layers.

## 1 Introduction

*In-context learning* (ICL) refers to the phenomenon in which large generative pretrained transformers (GPTs) perform tasks with no gradient updates when shown task examples or descriptions in their context [13, 12]. Recent work on in-context learning has focused on *prompt-engineering*, treating GPT models as black boxes and focusing on which examples to provide in-context [53, 14, 21, 22, 57]. However, many of these works apply surface level interventions leaving the internal mechanism of task recognition in GPT models largely not understood.

In this work, we ask **where does in-context Learning occur** in GPT models? Our view of In-context Learning is that of "task recognition" not "task learning" [69, 47]. While in-context learning in GPT models appears to be generally applicable to any natural language task, to study task location, we focus on two tasks, Machine Translation (MT) and Code generation, as there is little to no ambiguity in evaluating whether the model has recognized the task. For MT, the model must generate tokens in a different language. For Code generation, the model must produce a working program in the correct programming language. These two tasks are unlikely to be "learnt" from following patterns, and are more complex than a lookup in associative memory for simple Question-Answer tasks.

We focus on multi-head attention layers as a unit of study, as the self-attention mechanism is designed to allow the model to attend to it's context during generation of the target sentence [59]. Using causal masking over different parts of the context we demonstrate that there exists a "task-recognition" point after which attention to the context is no longer necessary (Section 4). Concurrent and previous

---

*Corresponding Author, suzyahyah@gmail.com and Code Repository https://github.com/suzyahyah/where_does_in-context-learning_happen_in_LLMs

work on Task and Function Vectors [32, 58] have also characterised a similar phenomena where the activations induced by in-context examples can be used to control tasks in the model.

We further characterise this phenomena, by studying the effect of on various ablations of masking self-attention over the instructions, examples, and even the query sentence itself. We report that not only is it *unnecessary to compute self-attention across the instructions and examples, in later layers self-attention over the query itself may also be redundant.*

This work informs the design of efficient inference and training for LLMs with the following contributions

1. We discover large computational savings when the context is several times longer than the test source sentence, a typical phenomena in prompt engineering (Section 5).
2. Parameter efficient fine-tuning corresponding to the phenomena of in-context learning. We observe that very lightweight fine-tuning of LoRA parameters [34] are most effective at earlier layers of the model compared to the later ones (Section 6). The effectiveness of the LoRA training corresponds directly to the layers that occur before the 'task recognition' point.

We further investigate the extent of MT *task redundancy* using differentiable $L_0$ regularisation to train discrete attention head gates (Section 5.1) and find that only around 10% of the attention heads can be fully masked. This indicates that the attention-heads themselves are not redundant, it is attention over all of the context that can be redundant. This fundamentally differs from the literature in supervised learning where more than half of the attention heads can be pruned, and Transformers are highly specialised for particular tasks [61, 44, 7].

## 2   Background

**In-Context Learning**   was first demonstrated by [13] who showed that GPT-3 could be used to perform a huge variety of tasks without any task-specific parameters or training, by conditioning the model's generation on a *prompt* which included a few labeled examples of the task of interest. Since then, interest in using GPT models for ICL has grown significantly [45, 3, 66], with several recent works introducing methods such as instruction-tuning [55, 63] or chain-of-thought prompting [64] to improve downstream ICL accuracy. One key characteristic of In-context Learning is its reliance on prompt examples demonstrating the task that the model should carry out [52].

**In-context Learning as Task Recognition.**   Ostensibly, ICL can work for nearly any task that can be defined or described in natural language, and therefore has potential for incredibly broad impact. However, ICL can often still underperform supervised fine-tuning [9], prompting research in analyzing the mechanisms underlying ICL. One line of work studies in-context learning with *linear* functions, typically linear regression, characterizing the learnability of these functions with ICL [39, 28] and even the learning algorithm a transformer uses [2, 18, 62]. A second body of work suggests that in-context learning locates *existing* latent concepts (tasks) which have been *already learnt* during pretraining [69, 65]. Notably, [58] describe function vectors which are robust to changes in context. [30] try to characterise the extent of task recognition from the pre-training data. Although there have been many studies on task recognition, our work presents a complementary perspective for task recognition, by demonstrating that there exists a point in the model's *layers* where the task has been located and causal self-attention onto the context is no longer needed for the model to perform the task.[2]

**Transformer Layers and Self-attention as the Unit of Study.**   Many works study layers of the model as a natural unit of analysis for interpretability [33, 20, 48, 24, 8, 54]. We highlight some of the work which is more closely related to task performance. [68] study the layer-wise adaptability by a hidden-state variability ratio while [60] study evolution of representations in MT-supervised transformer models. [49] studies when model layers can be skipped by feeding intermediate representations into the final output layer of a pre-trained supervised model. Our work

---

[2]In our experiments investigating where "task recognition" happens, we consider the actual "task performance" score as it could be possible to recognise the task as Machine Translation or Code generation, yet perform less well on it.

adds to this body of work by considering the perspective of when and where layers are responsible for task location in in-context learning models.

The self-attention mechanism specifically has been highlighted as a source of redundancy by many previous and concurrent works [10, 46, 31]. This is due to it's causal structure over the input symbols under the specific context of the input sequence within it's context window [51]. In this paper, we study a major source of causal redundancy in the input, the "prompt examples" that are provided as input-output demonstrations to the model for "in-context learning".

Transformer overparameterization and redundancy has been an active area of research [19] with multiple works suggesting to adapt transformer inference depth [36, 15, 25]. While we draw inspiration from these, our main objective is not to compress models for inference, but to highlight the redundancy in computing over long context token sequences.

## 3 Data and Settings

**Models** We use GPTNEO2.7B [11], BLOOM3B [56], LLAMA3.1-8B and LLAMA3.1-8B-Instruct in all of our experiments with Machine Translation. For code generation, we used LLAMA3.1-8B-Instruct[23] and STARCODER2-7B[41]. GPTNEO2.7B has 32 layers and 20 heads, BLOOM3B has 30 layers and 32 heads, LLAMA2-7B and LLAMA3.1-8B has 32 layers and 32 heads and STARCODER2 has 30 layers and 24 heads. The checkpoints we use are from Meta AI (for LLAMA) and the transformers library [67]. STARCODER2 and LLAMA models utilises grouped-query attention [1], while the rest of the models use "regular" multi-head self-attention.

GPTNEO was trained on The PILE [27], an 825GB text dataset which consists of roughly 98% English data. Despite being mostly monolingual, The PILE contains Europarl which GPTNEO was trained on at a document level (rather than a sentence level). Conversely, BLOOM was trained on the ROOTS corpus [38], a composite collection of 498 datasets that were explicitly selected to be multilingual, representing 46 natural languages and 13 programming languages. LLAMA training data consists primarily of common crawl, C4, wikipedia, stackexchange as major sources. STARCODER2 was trained on Github as well as Arxiv and Wikipedia. To our knowledge, there has not been any reports of sentence level parallel corpora in the training datasets of these models.

**Data** We test our models using two datasets, FLORES [29] for Translation and HUMANEVAL for Code generation. For FLORES, we experiment with en↔fr (main paper) and en→pt (appendix). Prompt examples are drawn from the development set. We evaluate the generations using BLEU scores, following the implementation from [50]. For HUMANEVAL[16], we evaluate on the execution accuracy of the generated code using the Pass@1 metric. As HUMANEVAL does not have an explicit train set, the prompt set is drawn from the Mostly Basic Python Program (MBPP) dataset [4]. To account for example selection and ordering effects,[3] all inference runs were repeated with 5 randomly sampled prompt example sets.

**Prompt Format** Our prompts may consist of instructions, examples, both, or none. Importantly, we adopt *neutral* delimiters, "Q:" and "A:" to separate the prompt and the start of machine generated text. This ensures that the models do not have any information from the delimiters on what the task is and must recognise the task from examples. [4]

For the translation task, when no natural language instructions are used the model input will be `Q: {source_sentence} A:` Instructions are given in natural language and take the form: `Translate from {L1} to {L2}: Q: {source_sentence} A:`, where `L1 = English` and `L2 = French` if the source and target languages are English and French respectively. Examples are given after instructions, and similarly delimited by Q: and A:. See Appendix: Table 1 for an example.

---

[3]In-context learning models have been found to be sensitive to these order effects [42].

[4]In an earlier exploration, we found that supplying the model with language indicators only, e.g., "English:", "French:" or "English:", "Python:", was sufficient for strong models (LLAMA3.1-8B and LLAMA3.1-8B-INSTRUCT) to perform the task without seeing any instructions or examples in the context.

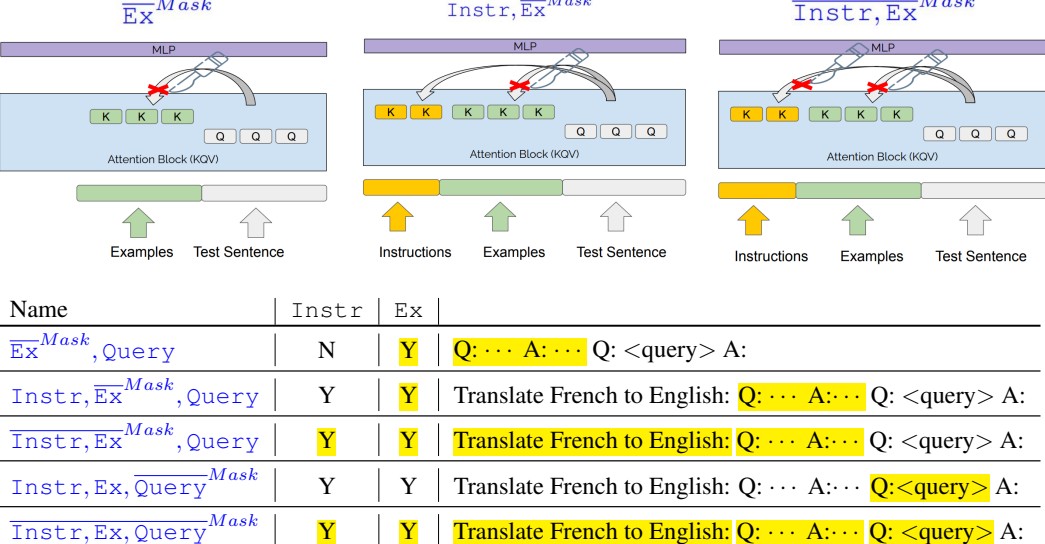

| Name | Instr | Ex | |
|---|---|---|---|
| $\overline{\texttt{Ex}}^{Mask}$,Query | N | **Y** | Q:⋯ A:⋯ Q: <query> A: |
| Instr,$\overline{\texttt{Ex}}^{Mask}$,Query | Y | **Y** | Translate French to English: Q:⋯ A:⋯ Q: <query> A: |
| $\overline{\texttt{Instr,Ex}}^{Mask}$,Query | **Y** | **Y** | Translate French to English: Q:⋯ A:⋯ Q: <query> A: |
| Instr,Ex,$\overline{\texttt{Query}}^{Mask}$ | Y | Y | Translate French to English: Q:⋯ A:⋯ Q:<query> A: |
| $\overline{\texttt{Instr,Ex,Query}}^{Mask}$ | **Y** | **Y** | Translate French to English: Q:⋯ A:⋯ Q: <query> A: |

Figure 1: **(Top):** Graphical explanation of Masking the Attention over Instructions and Examples. The leftmost image has instructions and masks examples ($\texttt{Instr},\overline{\texttt{Ex}}^{Mask}$), while the right image has both instructions and examples masked ($\overline{\texttt{Instr,Ex}}^{Mask}$). **(Bottom):** We demonstrate which components of the input prompt are masked for each setting that we experiment with. The overline of the setting name describes which portion of the input is highlighted (and thus masked). $N/Y$ refer to absence / presence of either Instruction ($\texttt{Instr}$) or Examples ($\texttt{Ex}$). Although we are primarily concerned with the effects of masking out task-identifying context (i.e. instructions and examples), in some experiments we additionally consider masking out the input query as well.

For the code generation task, when no natural language instructions are used, the model input will be $\texttt{Q: \{program\_description\}}$, where the $\texttt{program\_description}$ is Instructions are given in natural language and take the form: "Write a program for the following task:".

## 4 Where Does In-Context MT happen?

### 4.1 Analysis Methodology: Layer-from Masking

In-context learning differs from task-specific supervised learning in that, during test time, the desired task must be identified, or learned, from the context first and then applied to the input. At what stage in the feed-forward computation does a causal Large Language Model transition from an in-context learner to a translation or code-generation model? To explore this question, we introduce *layer-from context-masking* which masks out all attention weights to the context (instructions, examples, or queries) from a certain layer *onwards* (see Figure 1 for a graphical description).

For Causal Decoder-only Transformer Language Models, given each position $i$, the Attention weight $\alpha_{ij}$ over context positions $j, j < i$ can be computed by a $\alpha_{ij} = \text{softmax}(\frac{QK^T}{\sqrt{d_k}})_{ij}$. Each element in $(QK^T)$ is the dot product between a query vector and key vector $q_i \cdot k_j$, where $q_i = W_q x_i$, $k_j = W_k x_j$ for trained weight matrices $W_k$ and $W_q$.[5] We apply the attention mask over the input so that the attention score is $(q_i \cdot k_j) + m(j, \mathbf{u})$. Here, $\mathbf{u}$ are the tokens that we wish to mask, and

$$m(j, \mathbf{u}) = \begin{cases} 0 & \text{if } x_j \notin \mathbf{u} \\ -\infty & \text{if } x_j \in \mathbf{u} \end{cases} \quad \begin{array}{l} \texttt{layer.j.attn\_mask[i, :, :, u(start):u(end)] =} \\ \texttt{torch.finfo(attn\_mask.dtype).min} \end{array}$$

is implemented in practice as the smallest floating point value for that datatype. All masks operate from the $j$-th layer ($\ell_j$) *onwards*, i.e. masking from $\ell_{20}$ means zeroing out attention to all positions

---

[5]Readers should note that there is a $W_k$ and $W_q$ weight matrix for each layer and each attention head, but we omit the notation on this for readability.

in $\mathbf{u}$ from $\ell_{20:n_\ell}$, where $n_\ell$ is the total number of layers. To construct Fig 2, we increment $\ell_j$ from 1 to $n_\ell$ and apply the set of masks $\{m(j, \mathbf{u})\}^{\ell_j:n_\ell}$ in each experiment and observe the performance of the model.

When masking input tokens from layer $\ell$, the model must rely on only the information in the hidden state representations of the remaining, unmasked tokens from layer $\ell + 1$, since representations of the masked tokens can no longer be incorporated moving forwards; if the unmasked representations do not already encode enough information to complete the task (e.g., Machine translation) then the model will fail to generate the correct output. Our intuition is the following: if, at layer $\ell$, the model can perform the target-task without attending to the task *context*—task-identifying tokens such as instructions and examples—then information about the task has already been incorporated into the query representations and the model has identified, or "recognized", the target-task by layer $\ell$.

## 4.2 Experiments on *Layer-from context-masking*

In Figure 1 (Table) we show the various masking treatments that we apply to the input in our experiments. We ablate over 3 different task-context masking settings to test the impact of various parts of the context: providing only task examples and masking them from a given layer ($\overline{\texttt{Ex}}^{Mask}$, `Query`); including the instruction and but only masking out the examples (`Instr`, $\overline{\texttt{Ex}}^{Mask}$, `Query`); and including instructions but masking them with the examples ($\overline{\texttt{Instr}, \texttt{Ex}}^{Mask}$, `Query`). As a control in our experiments, we also experiment with masking the entire input ($\overline{\texttt{Instr}, \texttt{Ex}, \texttt{Query}}^{Mask}$) to study whether the model needs to attend to *any* input beyond a certain layer, and with masking only the query (`Instr`, `Ex`, $\overline{\texttt{Query}}^{Mask}$) to study whether masking the context vs the query have similar effects. For each masking setting, we apply the mask from all layers in the model ($j = 1, \cdots, n_\ell$) and observe how task performance is affected at each layer. When examples are provided in-context, we use 5 examples per prompt and we re-sample these examples to control for variance in example selection.

## 4.3 Results

**Models do not need to maintain attention over the task context past a certain layer to perform the task.** In all models, we observe that when applying masking from $\{m(j, \mathbf{u})\}^{\ell:n_\ell}$ over the task context, models obtain their maximum performance well before the final layer, i.e., when $\ell < n_\ell$. The results of our experiment for $\texttt{en} \rightarrow \texttt{fr}$ and $\texttt{fr} \rightarrow \texttt{en}$ are shown in Figure 2, and additional experiments for GPTNEO and BLOOM on $\texttt{en} \rightarrow \texttt{pt}$ and $\texttt{pt} \rightarrow \texttt{en}$ are shown in Section A.4. Different models reach this plateau point at different layers: in GPTNEO this point occurs around layer 25, in BLOOM this point occurs around layer 15-20, and in LLAMA models this occurs around layer 13-15. As English is the dominant language in most model's training, models can successfully perform translation into English upon earlier layers of masking, than translation out of English. Once this plateau is reached, the models benefits only marginally, if at all, from retaining attention to the context, suggesting most of the task "location" has already occurred.

We observed that with LLAMA-3.1 models, when masking only task context, there is a jump in performance from nearly negligible to nearly optimal in the course of a few layers. Conversely, when the query is masked we see both performance begin to rise much later in the model and the approach to optimal performance occur much more slowly, often plateauing only a few layers before the end of the model.

**In some models, there may be a point where forward computation is independent of even the query.** There *is* also a point in the model where it no longer needs access to any input query tokens. We find this effect to be much less pronounced in GPTNEO2.7B, BLOOM3B and STARCODER on the code generation task, and thus maybe a characteristic of Llama models training.

**There exists critical layers for task location**. Prior to the task recognition point, around the middle layers of the models, moving the context mask up a layer results in a significant increase to performance. We consider these critical layers, as instead of a gradual increase in performance, we observe very steep jumps, accounting for more than 80% of the model's ceiling performance for that task. We conjecture that the model is locating the correct task during processing in these middle layers, after which the context is no longer necessary to perform the task.

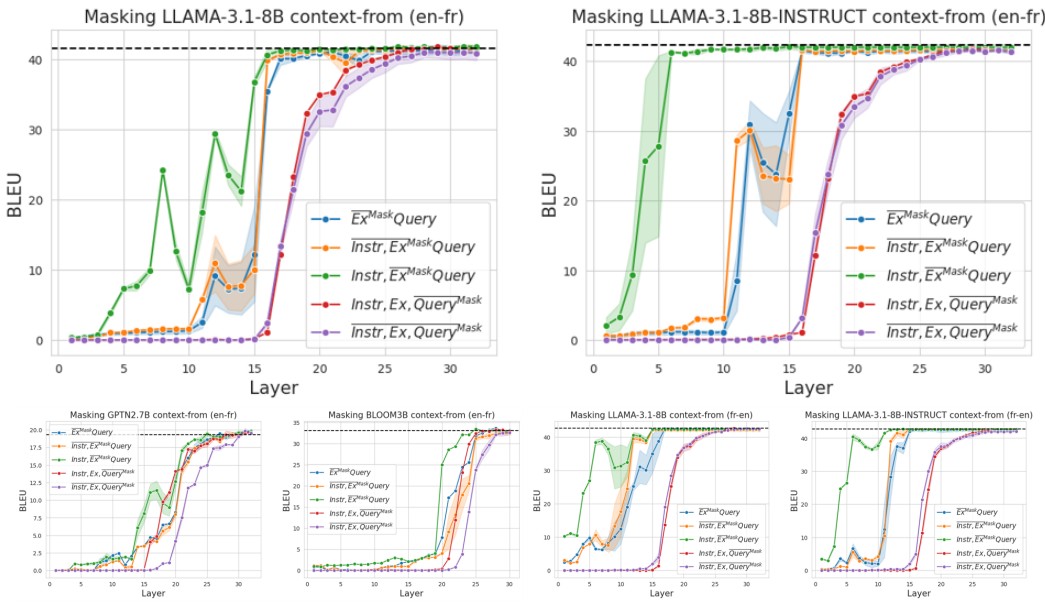

Figure 2: *Layer-from context-masking experiments* for LLAMA3.1-8B, LLAMA3.1-8B-INSTRUCT en→fr (main figure), and GPTNEO2.7B, BLOOM3B, en→fr, LLAMA3.1-8B, LLAMA3.1-8B-INSTRUCT on fr→en. The graphs show translation performance when masking contexts from the $j^{\text{th}}$ layer onwards. Different lines indicate different masking treatments, as described in Figure 1. The dashed black line is the performance when no masking of the input occurs.

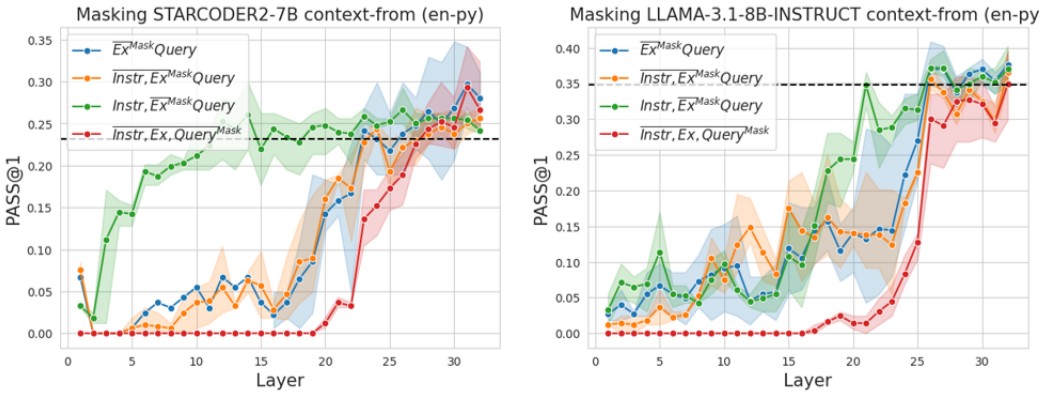

Figure 3: *Layer-from context-masking experiments* for Starcoder2-3B, Starcoder2-7B, Llama7b, Llama7b-chat on a text to code generation task. The graphs show translation performance when masking contexts from the $j^{\text{th}}$ layer onwards. Different lines indicate different treatments of the instruction, as described in Figure 1. The dashed black line is the performance when shown both examples and instructions without masking.

Overall, our findings suggest a 3-phase process to in-context learning: in the first phase, moving the mask up makes little difference in performance, which is close to 0. This suggests that the context has not influenced task location at all. In the second phase, shifting the mask upwards makes a large difference in performance, suggesting that the model has started to locate the task but can improve significantly with more processing of the context. Finally, in the third phase, shifting the mask upwards again has little-to-no effect on the performance, suggesting that the model has fully recognized the task as translation and no longer requires the context to interpret the task.

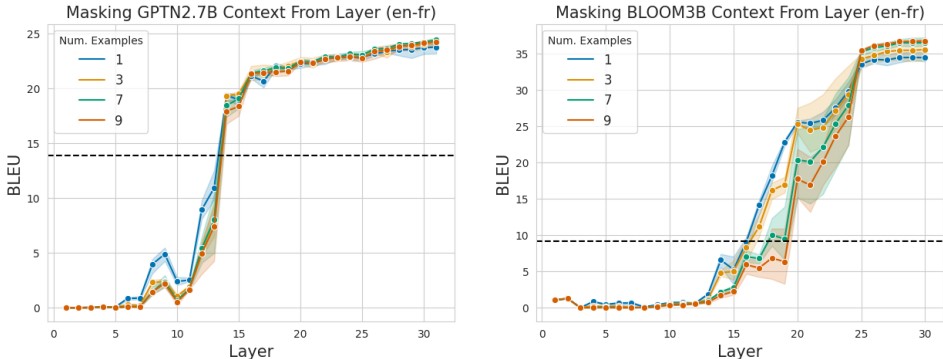

Figure 4: *Layer-from context-masking experiments* for GPTNeo and BLOOM on en → fr investigating number of examples in the $\overline{\text{Ex}}^{Mask}$ mask setting. The dashed black line refers to no instructions and no examples.

## 4.4 Instruction-tuned vs Non-instruction Tuned Models

When comparing non-instruction tuned LLAMA3.1-8B vs instruction-tuned models LLAMA3.1-8B-INSTRUCTION, we do not observe any noticeable difference in where performance plateaus, i.e., where the model no longer requires attention over the context. This occurs around layers 16 for both LLAMA models in en → fr and around layer 13 for fr → en. The main difference is that instruction-tuned model is able to achieve better performance in the earlier layers for the setting where instructions are present and examples are masked ($\text{Instr}, \overline{\text{Ex}}^{Mask}$). This is to be expected as these models are tuned towards following instructions.

Overall we find that the observation of task recognition layers and a task recognition point is present across both non-instruction tuned and instruction tuned models, and that this presents itself similarly in both types of models.

## 4.5 Do models have a distinct task recognition region regardless of the type of task? (Experiments on Code Generation)

For tasks that the model does not perform fluently, we do not observe a sharp increase at any particular layer. For instance, for code generation (HUMANEVAL) where the LLAMA2 model performs poorly, we can observe only a very gradual effect of masking the self-attention layers, and not a distinct increase as compared to the LLAMA2's performance on Translation.

However when we consider STARCODER2 while masking instructions or no instructions, i.e., the $\overline{\text{Instr}, \text{Ex}}^{Mask}$ and $\overline{\text{Ex}}^{Mask}$, we again see the same pattern demonstrating the task recognition phenomena on layer 19 of the 3B model, and layer 20-23 of the 7B model.

To understand Starcoder2's strong performance on the ($\text{Instr}, \overline{\text{Ex}}^{Mask}$) condition, investigations found that the instructions and the test prompt alone contain sufficient information for the model to recognise that the task is to generate a Python program, even though the model is not instruction tuned. This happens as the model is very specialised towards code generation and has a strong prior to generate python code given its prevalence in it's training data.

## 4.6 The Role of Instructions vs Examples

In separate experiments, we found that when shown only instructions and no examples, GPTNEO and BLOOM models are unable to translate, and their performance is nearly at 0 BLEU Score. For GPTNEO and BLOOM we see that the behavior of the model is similar when no instructions are present ($\overline{\text{Ex}}^{Mask}$) and when instructions are masked ($\overline{\text{Instr}, \text{Ex}}^{Mask}$). However, if the model is given complete access to instructions ($\text{Instr}\overline{\text{Ex}}^{Mask}$), it can use the intermediate processing of examples to reach baseline performance earlier.

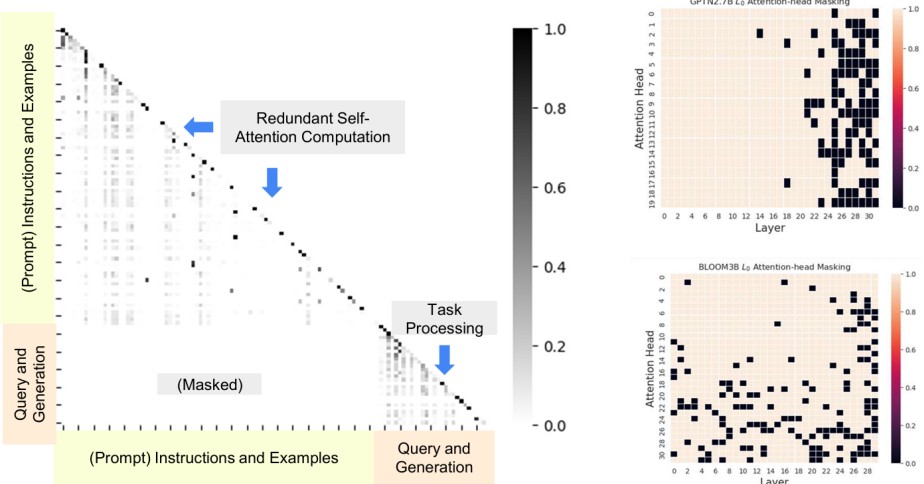

Figure 5: **(Left)** Illustration of redundancy in self-attention computation based on our masking setup ($\overline{\texttt{Instr},\texttt{Ex}}^{Mask}$, $\texttt{Query}$). **(Right)** Visualisation of attention head masks for GPTNEO and BLOOM, learned with $L_0(\lambda = 0.01)$ regularisation under a $\texttt{0-prompt train}$ scheme in $\texttt{en} \rightarrow \texttt{fr}$. A value of $0$ (in black) indicates that the attention head is effectively masked out by the trained attention gate. Around 10% of attention heads are masked out i.e., redundant, with a majority of them occuring at the later layers for GPTNeo and distributed across layers for BLOOM. $\texttt{fr} \rightarrow \texttt{en}$ is availble in Section A.7.1

## 4.7 Does the Number of Prompts Affect Task Recognition?

In Section 4 we study context-masking with a fixed number of prompts. However, it is not clear if the number of prompts affects how fast, layer-wise, the model is able to recognize the task. We plot these results for $\texttt{en} \rightarrow \texttt{fr}$ in Figure 4, for both GPTNEO and BLOOM. In general, we find that the number of prompt examples has little effect on which layer the task is recognized at. While there is some variation in performance when the context is masked around the middle layers of the model, the final performance plateau occurs at the same layer regardless of the number of prompts.

## 5 Inference Efficiency

Speeding up transformer inference is of great interest to the community [26]. We highlight the potential of speeding up inference time as a direct consequence of identifying where task recognition occurs in the model and redundancy of self-attention processing. In Figure 5, we illustrate self-attention from the query and generated sequence tokens over itself (Task Processing) and Self-attention over the prompt Instructions and Examples (Masked). If ceiling performance is achieved, then self-attention over the previous context becomes redundant (Redundant Self-attention Computation).

Our results indicate that we can achieve significant speedups in inference by removing the processing of context-tokens all-together after a certain point in the model, with little to no impact on downstream performance. Let $\ell_r$ be the $r^{\text{th}}$ layer where we can mask out the attention of the context across subsequent layers and match the "ceiling" performance. Let $k$ be the number of prompt examples, where each example consists of a pair of parallel sentences. Then, for a model with $n_\ell$ layers, the amount of processing in terms of speed and memory saved is approximately $(n_\ell - r)/n_\ell \times (k/k+1)$.

Using the example of LLAMA3.1-8B (32 layers) on $\texttt{en} \rightarrow \texttt{fr}$, we see from Figure 3 that the model is very close to it's ceiling score after processing the examples at layer 14 ($\ell = 14$). If we no longer need to process examples after $\ell = 14$, **under a prompt size of** $5$ **the savings are approximately 50%.**

For instruction-tuned models which are typically deployed in production, even if we assume that no examples are provided, savings can be non-trivial as very long-form instructions are typically provided to the model in an attempt to control it's behavior (prompt engineering).

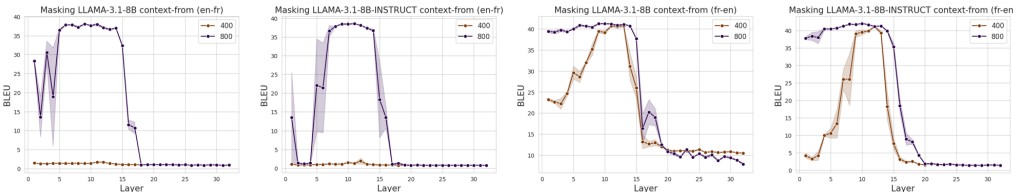

Figure 6: Performance of models ((LLAMA3.1-8B and LLAMA3.1-8B-INSTRUCT for en↔fr) trained with single LoRA layer, where each point on the x-axis reflects a single trained LoRA layer. The LoRA layer was trained without instructions, and with causal LM cross-entropy loss over next token prediction of parallel translation sentences. 400 and 800 refer to the size of the training set. The layers which are most amenable to lightweight fine-tuning occur in the earlier layers before the "task recognition" point.

Although we had demonstrated the redundancy of self-attention over the input context, the significance of this computational savings extends to *all* components of the transformer during forward inference. Since all subsequent layers of forward inference no longer rely on computations on previous token positions, all processing related to those redundant token positions (from the task recognition layer onwards) can be effectively removed.

Overall, observing redundancy over the context is not surprising. To explain why models can have such redundancy, we refer to [17] who identify a phenomena where attention heads attend almost exclusively to delimiter and separator tokens such as [SEP], periods and commas. This is thought to act as a "no-op" as the value of such tokens in changing the current hidden representation is very small. Note that it is then possible to mask entire Transformer layers and still achieve a sensible output due to residual connections in the Transformer architecture at every layer.

## 5.1 Are There Specialised Attention Heads?

A well established finding for supervised encoder-decoder MT models, is that up to 90% of the attention heads can be pruned while minimising fall in translation performance [61, 6, 44]. We note that asking about the extent of pruning is a slightly ill-formed research question, as it depends on the type of pruning technique used. However broad trends of highly prunable models have been observed in the supervised MT paradigm. For instance, [5] studied attention-head importance for a broader set of ICL tasks, finding that the most important heads for ICL occur in the middle layers of the model. We train discrete attention head gates with $L_0$ regularisation for GPTNEO and BLOOM on en → fr (see Section A.7.1). Overall, we report that there are no "few" specialised heads, which directly contrasts with the literature on compression in supervised MT models [61, 44]. Potential reasons for this difference might be due to cross-entropy loss associated with task tuning for MT vs non-specific training on large corpora. We leave this as an open question for future work.

## 6 The Adaptability of Task Layers

The layers prior to "task recognition" should contain information about locating the MT task. To test this, we further explore the adaptability of these layers by lightweight fine-tuning experiments on LLAMA3.1-8B and LLAMA3.1-8B-INSTRUCT on en↔fr.

We trained a single Low-rank Adaptation matrix (LoRA; [34]) for each layer of the output projection while keeping the rest of the network frozen.[6] This means there were $n_\ell$ individual layers trained for $n_\ell$ experiments in Figure 6, where $n_\ell$ is the total number of layers of the model.

The model was shown parallel sentences as input, and layers were trained with no explicit translation instructions. We split the dev set of FLORES into 400 and 800 training examples and 200 dev examples, we repeated the experiments with 2 random seeds initialisations. Note that this setup is designed to tune the layers for task location. It is highly unlikely that the model can learn translation

---

[6]We also experimented with the training separate Key, Query and Value LoRA Layers but found this to be less effective.

knowledge from this small amount of supervision. The LoRA layers were trained for up to 50 epochs with batch size= 32, learning rate= $1e-4$, early stopping patience= 5 and threshold= 0.01, with $\alpha = 32, r = 8$ and dropout= 0.05. These values are default and there was no hyper-parameter optimisation over the training parameters. The cross-entropy loss was computed across the entire sequence, and we used the best checkpoint on the 200 held out dev examples for evaluation.

Without any fine-tuning, performance is close to 0 because the model will generate sequences continuing from the source language instead of doing translation. **While each layer can be trained to perform better than no fine-tuning at all, tuning different layers have vastly different impacts on performance** (see Figure 6). In particular, we find that high performing layers occur at the earlier to middle parts of the network, with the peak occuring strictly before the "task-locating" layers from Section 4.

### 6.1 Task Locating Layers are critical for resource efficient fine-tuning

With half the number of training examples (400 instead of 800), the range of trainable layers drop very greatly. For the more challenging direction of generating in French, en → fr, reducing the number of training examples result in none of the layers being successfully fine-tuned for translation task location. For fr → en, the range of trainable layers is much more concentrated around layers 10 to 15, which occurs just before the 'task recognition' layers as shown in Figure 2.

We demonstrate that in contrast to common fine-tuning wisdom, additional tuning on later layers in the transformer network has a much smaller impact on final performance, and this is strong correlated with where the 'task locating' layers are in the model. The major reason for this discrepancy from conventional fine-tuning wisdom, is that we are performing extremely lightweight parameter-efficient fine-tuning for task location, and not full fine-tuning on large datasets. Our results should thus be interpreted under the lens of highly resource efficient fine-tuning of layers for task location, and is fundamentally different from the wisdom of "true" task fine-tuning.

## 7 Conclusion

We demonstrate evidence that In-context Causal Decoder models locate their task at a specific layers during forward inference. To study this, we introduced causal masking of self-attention over the context from layer $\ell$ onwards (Section 4). The findings generalise across 4 models of different sizes and in both non instruction-tuned and instruction-tuned models. We further identify certain layers as task critical, and show that this corresponds to the task recognition point of the model (Section A.9) and is not influenced by increasing number of examples (Section 4.7) shown to the models.

Our central finding that models do not need to maintain attention over all of the context across every layer has direct implications for inference efficiency of transformers, with estimated up to 45% cost-savings for llama model with 5 examples (Section 5).

Contrary to common fine-tuning wisdom, we show that it is sometimes beneficial to target middle layers for fine-tuning the model which could be associated with task recognition ( Section 6). Finally, we trained attention head gates using differentiable $L_0$ regularisation (Section 5.1), and found that around 10% of attention heads can be masked. These are mostly distributed across the later layers of the model, providing some support for the idea that later layers are redundant. Although we have characterised this phenomena using Machine Translation and Code Generation, we believe that the broad findings are likely to generalise to other tasks.

### 7.1 Limitations (and Future Work)

- There is limited exploration of why different models exhibit varying behaviors in terms of their "task recognition point" and critical layers. Unfortunately, the differences are not due to easily observable hyperparameters like model size or architecture. To put in another way, why do large models exhibit different characteristics?

- This paper reports on empirical analysis and observations, and currently lacks a more theoretical framework that could explain why this phenomena is being observed.

**Acknowledgments**

We would like to thank all the anonymous reviewers for their invaluable comments and suggestions, as well as Daniel Kashabi and Marc Marone for feedback on earlier drafts.

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

# A  Appendix

## A.1  Reproducibility

The Machine Translation dataset that we use, FLORES [29] and the code generation datasets HumanEval[16] are fully open-source and well-known in the community. All models are open-source and freely available on Huggingface [67]. We used models of "reasonable" size (3B and 7B parameters) that can be run with consumer or academic grade GPUs, making our work reproducible to most academic institutions.

## A.2  Impact Statement (Ethics and Societal Consequences)

There are no known ethical concerns as these are exploratory studies on open-source LLMs.

## A.3  Prompt Format

| Translate English to French. | | |
|---|---|---|
| Q: A discomfort which lasts .. | A: | Un malaise qui dure |
| Q: HTML is a language for formatting | A: | HTML est un langage de formatage |
| ... | | ... |
| Q: After you become comfortable with formatting .. | A: | |

Table 1: A single continuous input sequence presented to the model for decoding a single test source sentence "After you become comfortable with formatting..". Given the entire sequence as input, the model proceeds to generate the target sequence.

## A.4  Additional Results on English & Portugese

In addition to the language pairs en $\rightarrow$ fr and fr $\rightarrow$ en, we also run experiments on English and Portugese language pairs, for en $\rightarrow$ pt. Due to space limitations, we plot the results of those experiments here. Overall, we see largely identical trends on both directions of English and Portugese to what we observe on English and French translation tasks, leading us to conclude that our conclusions generalize across different translation tasks.

## A.5  Autoregressive Decoder only Transformer

The transformer consists of stacked blocks of self-attention, which itself consists of smaller units of self-attention heads that are concatenated before being fed through a fully connected layer. In autoregressive decoder-only transformers, training and inference adopts a causal mask, where current positions are only able to attend to previous timesteps, instead of being able to attend to the entire input sequence. Unlike encoder-decoder NMT models where source and target sentence have separate processing transformer blocks, decoder-only means that the same model weights are both used to "encode" the source sentence and "decode" the target sentence in a single continuous sequence.

## A.6  Training with Autoregressive Translation

The original language modeling objective in GPT training involves predicting the entire input token sequence which consists of both the source and target sentence (shifted by 1 position). We found this to produce slightly worse results than only minimising the negative log likelihood of predicting the target sentence to be translated, and not the entire sequence. We consider this autoregressive translation training.

## A.7  $L_0$ Attention Gate Training

**Training Details**  For Section A.8, We train using Adam Optimizer ($\beta_1 = 0.9, \beta_2 = 0.999$) with a batch size of 32, and learning rate of 0.001, early stopping patience of 10 and threshold of 0.01. We initialise attention head gates to be 1 instead of random or 0.5 as this leads to faster convergence. We experiment with two different training settings, the `0-prompts Train` setting

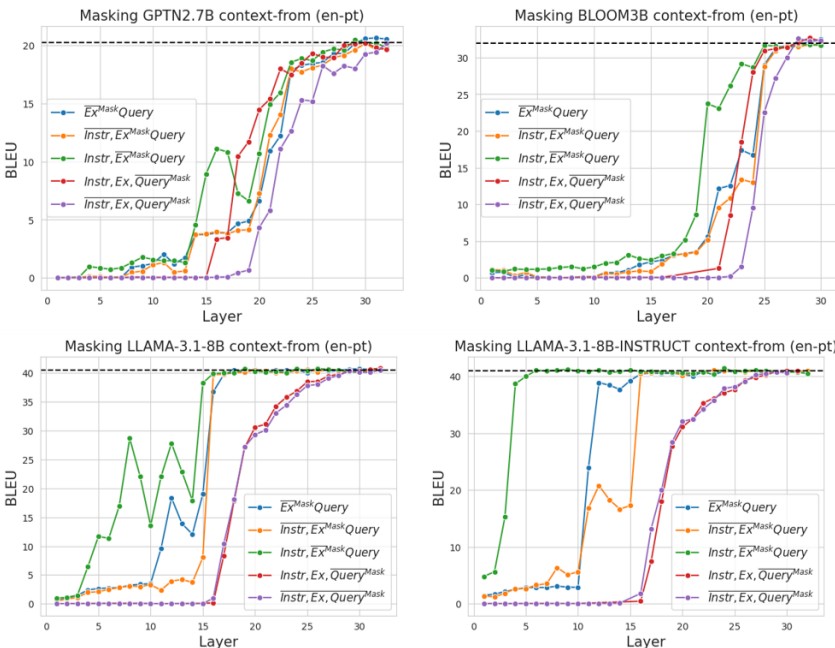

Figure 7: Layer-wise Context-masking results on the **English → Portugese** translation task. Critically, we see nearly identical trends to what we see in Figure 2 on the English to French translation task, suggesting our results generalize across language pairs and "tasks".

and the `5-prompts Train` setting. As described in Section A.6, we train the model by predicting only the target sentence, conditioned on the context. In the 0-prompt setting, the context consists of the instructions and the source sentence to be translated. In the 5-prompt setting, the context consists of the instructions, 5 prompt examples, and the source sentence to be translated.

In the `0-prompt` setting, the conditional prefix consists of the instructions and the source sentence to be translated. In the `5-prompt setting`, the conditional prefix consists of the instruction, 5 source target sentence pairs, and the source sentence to be translated.

**Data** We used the first 10,000 lines of en→fr from WMT06 Europarl [37] for training.[7] To test the generalisability of trained attention head gates, we use a different test domain, FLORES [29] to reflect the scarcity of in-domain data. We also test an additional language direction en→pt in FLORES to see if training can generalise across languages.

**Training Details** We train using Adam Optimizer ($\beta_1 = 0.9, \beta_2 = 0.999$) with a batch size of 32, and learning rate of 0.001. We use a large early stopping patience of 10 and threshold of 0.01, and train for up to 100 epochs. This is due to the nature of $L_0$ training; we do not expect performance to improve over many iterations and would like the attention gates to keep training as long as there is no large loss in performance. We initialise attention head gates to be 1 instead of random or 0.5 as this leads to much faster convergence and better performance. For the regularisation weight $\lambda$, we search over a hyperparameter set of $\{0.1, 0.01, 0.001, 0.0001\}$ and found $0.01$ performs best on the validation set.

### A.7.1 $L_0$ **head masking experiments.**

Additional experiments on L0 head masking in the en→ fr direction.

---

[7]Data available from `https://www.statmt.org/europarl/`

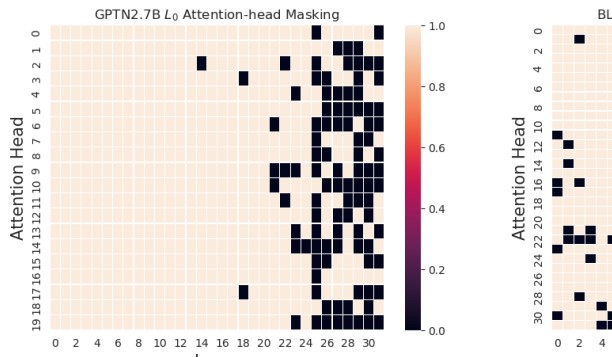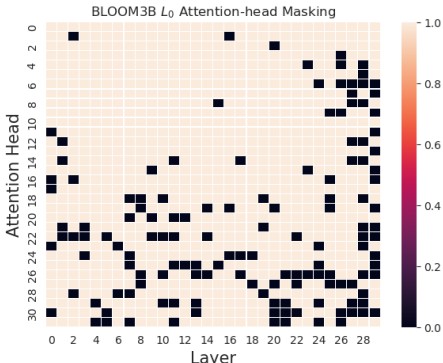

Figure 8: Visualisation of attention head masks for GPTNeo and BLOOM, learned with $L_0(\lambda = 0.01)$ regularisation under a `0-prompt train` scheme in en → fr. A value of 0 (in black) indicates that the attention head is effectively masked out by the trained attention gate. Around 10% of attention heads are masked out i.e., redundant, with a majority of them occuring at the later layers for GPTNeo and distributed across layers for BLOOM. fr → en is availble in Section A.7.1

### A.7.2 Training Attention Head Gates with $L_0$ regularisation

For a scalable approach to pruning, we opt to train self-attention head gates following [61] using the technique of differentiable $L_0$ regularization [40]. Let the attention head gates $g \in \mathbb{R}^{n_h \times n_\ell}$ be a set of trainable parameters, where $n_h$ is the number of attention heads per layer, and $n_\ell$ is the number of layers. Let the original output of each attention head be $v_j$, gated outputs $\tilde{v}_j$ are obtained by elementwise multiplication of the gate value $g_j$, i.e., $\tilde{v}_j = g_j \odot v_j$. For $\{(x, y)\}^n$ source sentence ($x$) and target sentence ($y$) training pairs, a model $f$ and loss function $\mathcal{L}$, $L_p$ regularisation adds a $\lambda$ weighted penalty associated with the complexity of the parameters. [8] The $L_0$ loss is non-differentiable as it involves raw counts of parameters. As a work around, $g$ can be approximated with random variables drawn from a Binary concrete distribution [43, 35].[9] We refer the reader to [40] for the relevant technical exposition. Details of training are provided in Section A.7.

### A.7.3 Generalisability of $L_0$ gate training

We experiment with `0-prompts` and `5-prompts` in training and using $\lambda = 0$ (no regularisation) and $\lambda = 0.01$. $L_0$ training for the `0-prompts` shows some gains for the 0-prompts test case, and with no loss on the 5-prompts test case (Table 2). Notably, this persists in en → pt, a different language direction from training.

The robustness of translation performance under multiple testing conditions (number of prompts, datasets, language directions) gives some confidence that the trained discrete attention head gates from $L_0$ support a general ability to translate (Table 2). In contrast, the soft attention head gates without regularisation ($\lambda = 0$) appear to overfit as they perform well on some conditions but deteriorate in others.

We observe that `0-prompt` training for $L_0(\lambda = 0.01)$ also outperforms `5-prompts` which is slightly suprising since `5-prompts` has more information in the prefix to locate the translation task. One possibility is that the model overfit to the Europarl domain where the training prompts were drawn from.

### A.8 Studying Redundancy via Compression

(Note: This Appendix section is based on LLAMA2-7B models).

---

[8]$L_2$ regularisation has the effect of reducing the magnitude of all $g$, $L_1$ regularisation has the effect of reducing the magnitude of several attention heads to a very small value (but not exactly 0), while $L_0$ regularisation has the effect of driving $g$ values to exactly 0.

[9]The class of Concrete distributions was invented to work around the problem of automatic differentiation of stochastic computation graphs.

| | Base | 0-prompts | | 5-prompts | | Base | 0-prompts | | 5-prompts | |
|---|---|---|---|---|---|---|---|---|---|---|
| | | $\lambda=0$ | $\lambda=.01$ | $\lambda=0$ | $\lambda=.01$ | | $\lambda=0$ | $\lambda=.01$ | $\lambda=0$ | $\lambda=.01$ |
| 0-prompts | 18.3 | 21.4 | 20.1 | 18.9 | 19.3 | 6.7 | 15.7 | 8.6 | 13.2 | 6.4 |
| 5-prompts | 24.3 | 24.5 | 24.1 | 23.6 | 24.2 | 25.9 | 19.6 | 25.8 | 24.3 | 26.0 |
| | Train: en→fr, Test: en→fr | | | | | Train: en→fr, Test: en→pt | | | | |

Table 2: Performance when using trained attention head gates for $L_0$ with regularisation $\lambda = .01$. $\lambda = 0$ refers to training without regularisation. 0 and 5 prompts were used in the context for training. We highlight values which are `greater` or `worse` than 0.5 BLEU points from baseline. Note that as these are compression experiments, we do not expect $L_0$ to perform better than baseline.

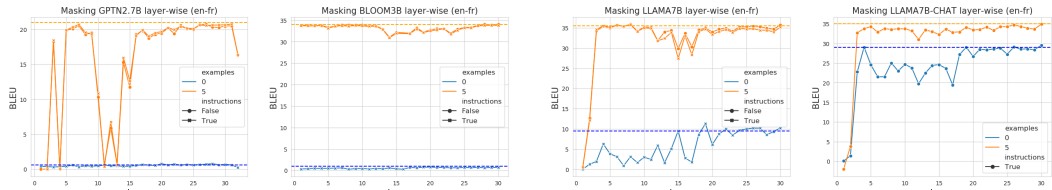

Figure 9: *Layer-wise masking* of self-attention heads for GPTNEO2.7B, BLOOM3B, LLAMA and LLAMA-CHAT on en → $fr$. The orange and blue dotted lines refer to the baselines (without masking) of 0 and 5 prompts with instructions. True/False refers to whether there are instructions provided (True) vs not provided (False). We observe critical layers near the middle and redundant layers towards the end of the model.

To what extent are there specialised attention heads for MT in the GPT-style models? If there were specialised heads, we would expect the model to be highly compressable/prunable to a select few heads. We plot a grid map of learned attention gate values for en → fr, where 0 indicates that the head is masked out (Figure 8). We find that most of the masked heads are distributed at the later layers for GPTNeo and are distributed across layers for BLOOM.

## A.9 Characterising Redundancy in Layers

Recently, [54] found that many layers in pre-trained transformers can be dropped with little harm to downstream tasks; moreover, it is well known neural MT transformer models are known have several redundant heads which are not necessary during test time [61, 44, 7]. However, it is not clear if the same trends hold for *in-context MT* models, and how that redundancy is related to task location versus task execution. We focus on the task of en → fr in this set of experiments.

We study the contributions of individual attention-layers by performing a simple *layer-wise* masking of all self-attention heads for a single layer. When we mask layer $j$, we are masking the *attention mechanism* of layer $j$, that is the MLP of layer $j$ acts directly on the output of layer $j-1$, rather than the output of the attention-head of layer $j$. Doing so allows us to study how *critical* each layer is, where *critical layers* is loosely defined as those that have a large negative impact when masked.

We plot results for each layer all models, using three combinations of {0 examples, no instructions}, {5 examples, instructions}, {5 examples, no instructions} in Figure 9.[10]

In Section 4, we observed that there are layers for task location. In this section, we observe evidence that there are critical layers which correspond to the task locating layers, providing support for our earlier observations.

For instance for LLAMA2-7B en → fr, even in the scenarios when examples are provided, we can see a dip in performance around layer 15 to 18. Refering back to Figure 2, we see that this is where most of the task location with large jumps in performance had occurred.

---

[10]The combination of {0 examples, no instructions} is not meaningful as the model only receives "Q: <source sentence> A:" as the input and is not expected to do the translation task.

For GPTNeo, we obseve a large set of contiguous layers which significantly decrease performance at around layer 10 to 15. This also corresponds to where most of the task location (large jumps in performance) had occurred for this model in Figure 2.

We note that the critical layers in different models have varying degrees of severity. It is not immediately clear why GPTNEO has such critical layers and suffers compared to the other models, although we note that this is unlikely to be due to size or model architecture as BLOOM is also around the same size as GPTNEO and performs more similarly to LLAMA. We suspect that it could be due to training data or some other factor related to the training dynamics but leave this for future work.

With regard to redundancy, we find that layers can be more safely removed towards the end without a noticeable loss in performance. We observe that for the less stable models, the model achieves close to baseline performance by layer-wise masking from $\ell_{15}$ for GPTNEO, $\ell_{26}$ for BLOOM and $\ell_{20}$ for LLAMA. This suggests that these later layers contain redundancy for translation.

