# OpenReview forum: "Where does In-context Learning Happen in Large Language Models?"
_NeurIPS.cc/2024/Conference — NeurIPS 2024 poster_

### Official Review · Reviewer_z6j9 · 2024-06-30

**Soundness:** 3
**Presentation:** 3
**Contribution:** 1
**Rating:** 4
**Confidence:** 4

**Summary:**

This study primarily investigates where in-context learning occurs within GPT-style models. Specifically, it explores the stage at which a model transitions from functioning as an in-context learner to a task-specific model. By applying layer masking to the instruction and in-context examples in machine translation and coding tasks, the authors observe performance changes to understand the internal mechanisms of in-context learning. They find that certain critical layers within the model are crucial for in-context learning.

**Strengths:**

The experimental design is well-conceived and includes an analysis that elucidates the internal mechanisms of in-context learning.

**Weaknesses:**

1. The practical utility of the findings is questionable. I doubt whether the proposed improvements to inference efficiency in Section 5 can be applied in practice. For example, critical layers differ across tasks, and even within the same task, they can vary between subtasks (e.g., en→fr vs. fr→en). Identifying critical layers typically requires significant cost.

2. The experiments are entirely based on multi-head Attention, which limits the applicability of these findings. Most current models use other attention methods, such as the grouped-query attention used in Llama-2. The findings in this paper may not apply to new attention variants, which already consider partial context attention.

**Questions:**

Please refer to the weaknesses section.

**Limitations:**

--

---

> ### Author Rebuttal · Authors · 2024-08-05
>
> We would like to thank the reviewer for their time and comments on our work. Regarding the listed weaknesses of our paper:
>
> > "The practical utility of the findings is questionable."
>
> **We respectfully disagree that our findings hold little practical utility. LLMs are increasingly being adapted to, and used as, task-specific models;** For instance, LLMs have shown great potential in replacing traditional supervised Machine Translation models. Our results hold incredibly high practical utility for scenarios where we require an LLM to adapt to a specific task using in-context examples and prompts, which will be used for every forward-pass of the model. As the length of the in-context examples increases (which we show does not significantly alter the position of critical layers), our results become even more relevant.
>
> > doubt whether the proposed improvements to inference efficiency in Section 5 can be applied in practice.
>
> While the exact position of critical layers is task-specific, finding them requires a one-time cost using only forward-passes of the model (e.g. no gradients or backpropagation is used). This is quite straightforward to implement, is highly reproducible, and requires no additional memory beyond forward inference.
>
>
> > The experiments are entirely based on multi-head Attention, which limits the applicability of these findings. Most current models use other attention methods, such as the grouped-query attention used in Llama-2. The findings in this paper may not apply to new attention variants, which already consider partial context attention.
>
> **We do in fact use Llama-2** in our experiments although we did not denote this in the paper, as it was the default Llama family available through hugging face at the time. Specifically, the checkpoints we use are https://huggingface.co/meta-llama/Llama-2-7b-hf and https://huggingface.co/meta-llama/Llama-2-7b-chat-hf. We sincerely apologise for the confusion.
>
> However, we also note that (to the best of our knowledge) Llama-2 uses standard multi-head attention. _Grouped query attention (GQA) is used in the more recent Llama-3, which was released only a month ago and is therefore out of the scope of our submission._ We analyze a broad set of highly utilized LLMs which suggests that our findings are broadly relevant to the LLM community. Our overall findings (that some layers are critical to in-context learning and that attention to the entire context is no longer necessary past a critical point) are also not specific to any particular form of attention.

---

> > ### Comment · Reviewer_z6j9 · 2024-08-12
> >
> > Thank you for your reply. Let me clarify that in my previous response, I mistakenly wrote llama-3 as llama-2. As you mentioned in your reply, this method need one forward-pass for each task, which makes it practical only when there are many requests for the same task. But that's not the case when we deploy our model in reality. In practice, we receive diverse requests from different users.

---

> > > ### Author Response · Authors · 2024-08-12
> > >
> > > Thank you for clarifying and sharing your expected scenario. We would like to address this scenario, and your comments around it:
> > >
> > > > this method need one forward-pass for each task, which makes it practical only when there are many requests for the same task. But that's not the case when we deploy our model in reality. In practice, we receive diverse requests from different users.
> > >
> > > The scenario we describe of adapting a general purpose LLM to a specific task, after which the LLM processes many requests for the same task, is an increasingly popular technique. For instance, in both research [1] and industry [2], there has been a shift towards using prompted LLMs to handle machine translation, under which a fixed prompt and model is used to handle many requests for a single task (i.e. translation from one specified language to another). This is precisely a setting that we study, and we therefore argue our analysis has very real practical implications for these techniques.
> > >
> > > Thank you again for your continued engagement with us!
> > >
> > > [1] A Paradigm Shift: The Future of Machine Translation Lies with Large Language Models,  Lyu et al., 2024
> > > [2] https://www.welocalize.com/insights/google-teams-up-with-welocalize-to-test-its-adaptive-translation-llm-solution/

---

> ### Author Response · Authors · 2024-08-11
> **Review clarifications**
>
> Dear reviewer, you raised two weakness which we provided responses to... we hope you might reconsider your overall assessment about the contributions of the paper.

---

### Official Review · Reviewer_m9xg · 2024-07-08

**Soundness:** 1
**Presentation:** 3
**Contribution:** 3
**Rating:** 3
**Confidence:** 4

**Summary:**

This paper tries to locate where LLMs handle in-context learning (ICL) and interprets how LLMs perform ICL internally. The authors propose a layer-wise attention masking strategy and conclude that LLMs perform ICL in bottom layers.

**Strengths:**

The motivation of this paper is clear, and the research question sounds interesting.

**Weaknesses:**

However, I have several concerns about the work.
1. Layer-wise masking. The authors only release first j bottom layers and conclude that bottom layers are important. This is problematic. If the authors mask layers with a reversed order, I suppose the conclusion would change to that top layers are important. The reason is that pretrained LLMs solve task with multiple modules. And different layers may share similar functions for solving that task. Simply masking several layers or layers with specific order cannot prove whether these layers are important/useless. It’s similar to the glove game: you have 3 gloves with 2 left gloves and 1 right glove. Dropping 1 left glove randomly cannot prove this dropped one is useless, or conclude that the right one is the only necessary one.
2. Token masking. The three token masking strategies are not comprehensive. There is no discussion about the individual examples. Either dropping all of them or keeping all of them cannot explain how ICL works, since many existing works show that ICL examples affect the performance much.

Overall, I think this paper tries to explore an important research question, and I would like to encourage the authors to address my two concerns.

**Questions:**

Line 117: can j be equal to I, i.e., (j<=i)?

**Limitations:**

See weakness.

---

> ### Author Rebuttal · Authors · 2024-08-05
>
> We would like to thank the reviewer for their time and comments on our work. Regarding the listed weaknesses and questions:
>
> > Layer-wise masking. The authors only release first j bottom layers and conclude that bottom layers are important.
>
> **Our conclusion is _not_ that bottom layers are important and top layers are not, but rather that there are critical layers.** We write that “Doing so allows us to study how critical each layer is, where critical layers is loosely defined as those that have a large negative impact when masked.. (line 231-232) ”.
>
> We explicitly test the relevance of individual layers across the _entire_ model. Showing here that dropping an individual layer (layer-wise masking) significantly harms performance does suggest that this layer is important because performance would not suffer if its functionality was re-implemented by a different layer.
>
> > If the authors mask layers with a reversed order, I suppose the conclusion would change to that top layers are important.
>
> The reviewer may be referring to _layer-from-context masking_ (Figure 2,3),
>
> The masking order that we have used, retains the original order of processing that the transformer sees during training.
>
> "All masks operate from the j-th layer ($\ell_j$ ) onwards, i.e. masking from $\ell_{20}$ means causally masking
> out attention to all context positions from $\ell_{20:n_\ell}$, where $n_\ell$ is the total number of layers. To construct
> Fig 2, we increment $\ell$ from 1 to $n_\ell$ and apply the set of masks $\{m(j, u)\}^{\ell:n_\ell}$ in each experiment and observe the performance of the model" (lines 122:125)
>
> The reverse order would mean skipping several layers of attention before passing the representations to the later layers, which may result in token representations that fundamentally differ from what the later layers have seen during training. It would not be clear whether performance was impacted because the context was masked until that point, or because the masked token representations no longer match the expected representation distribution at that layer.
>
> > pretrained LLMs solve task with multiple modules. And different layers may share similar functions for solving that task.
>
> We do not present any findings in this paper that conflict with this interpretation! In our masking experiments, we only mask over the context, not the entire test input provided to the LLM. The goal of the paper is to demonstrate **where in-context learning (in-context task recognition) happens**, not where the LLM "solves the task". (Line 130) If this is unclear, we hope to be able to clarify.
>
>
>
> > Token masking. The three token masking strategies are not comprehensive.
>
> To recap, the three token masking strategies are
> * Has no instructions, and masks examples ($\overline{\texttt{Ex}}^{Mask}$)
> * Has instructions and masks examples ($\texttt{Instr}\overline{\texttt{Ex}}^{Mask}$)
> * Both instructions and examples masked ($\overline{\texttt{Instr}}\overline{\texttt{Ex}}^{Mask}$) -- “dropping all of them”
>
> We described the significance of $\overline{\texttt{Instr}}\overline{\texttt{Ex}}^{Mask}$ i.e., “dropping all of them” in Section 4.1, Layer-from Context masking.
>
> "Under this causal masking treatment masking from layer ℓ, the model must rely on the representations of the target input sentence from layer ℓ + 1 only to complete the task; if the target sentence representations do not already encode the target task (translation into a specific language) then the model will fail to generate translations." (line 126:129)
>
> To put in another way, “Dropping all of them” and showing that performance continues to be maintained past a certain layer, demonstrates that the major work of in-context learning has occurred before that layer (Figure 2, 3). This finding is extended to a varying number of In-context examples (Figure 6).
>
>
> > many existing works show that ICL examples affect the performance much.
>
>  We agree that ICL examples are known to affect performance. Our experimental results are averaged over 5 random seeds (random in-context samples are used), to present findings which generalise beyond the choice of specific examples.
>
> > Line 117: can j be equal to I, i.e., (j<=i)?
>
> Yes, thank you for catching this typo!

---

> > ### Author Response · Authors · 2024-08-11
> > **Review clarifications**
> >
> > Dear reviewer, if our response has helped to clarify the misunderstandings, we greatly appreciate if you could reconsider your assessment!

---

> > ### Comment · Reviewer_m9xg · 2024-08-13
> > **Reply**
> >
> > Thanks for the reply. However, my concerns are not well addressed in the reply: the layer-masking strategy is problematic, and thus the following obtained conclusions are not reliable to me. Therefore, I'll keep my score.

---

### Official Review · Reviewer_NuLE · 2024-07-12

**Soundness:** 4
**Presentation:** 3
**Contribution:** 4
**Rating:** 8
**Confidence:** 4

**Summary:**

In-context learning has emerged as an important paradigm in LLMs. In this paper, the authors attempt to characterize where models learn to “recognize” an in-context task.

To do this, in-context portions (examples and/or instructions) are masked out after certain layers and are not included to generate model predictions.  An advantage of this style of masking is improved cost. Their experiments give clear results, indicating that for each of the selected tasks (machine translation, code generation), there exists a cutoff layer beyond which masking is acceptable without hindering ability to recognize the task. The most clear utility of this masking is computational savings. The authors also find evidence of 3-phase in-context learning, with the last phase accounting for little to no performance improvements.

**Strengths:**

* This is an important area of research. The authors carve out a strong motivation, which is enhanced by clearly defined and executed experiments.
* The paper is very well written, which makes it an interesting read.
* The experiments are extremely thorough, some of them being: ex v/s Inst masking, MT v/s code, attention to context v/s input, # prompts analysis, attention heads study etc. I can tell that these took a lot of effort and will be of great importance to the community.

**Weaknesses:**

I have some concerns about the definitions of “task recognition” [more in the Questions], as well as explanations on some sections (like Sec 4.3 and 6.2). It would be important to iron these out.

**Questions:**

It's not clear why “task recognition” v/s actual task performance was treated as a metric here. It seems that the learnings would be very different between the two. Is task recognition in itself important enough to celebrate over computational wins? Some clarity would be good in the paper to ensure readers understand the importance of "task recognition". Also, is there a cutoff (like BLEU > 0 means task recognition)?

Sec 4.3 could use better pointers. It's not clear which model is instruction tuned v/s not.

Sec 6.2 presents an interesting experiment; but the figure needs to be improved. The explanations don’t match the legends of the figure (for instance, what is true/false). Also, do we mask the source sentence as well?

---

> ### Author Rebuttal · Authors · 2024-08-07
>
> Thank you for the deep appreciation of the work!
>
> >  It's not clear why “task recognition” v/s actual task performance was treated as a metric here. It seems that the learnings would be very different between the two. Is task recognition in itself important enough to celebrate over computational wins? Some clarity would be good in the paper to ensure readers understand the importance of "task recognition". Also, is there a cutoff (like BLEU > 0 means task recognition)?
>
> It’s a fair point. The process of task recognition, happens over several layers. We think task performance as a metric is easier for people to understand.. Because finally the practical takeaway is that you can reach the model's ceiling task performance without needing full-processing over the context - that’s the computational win.
>
> > Sec 4.3 could use better pointers. It's not clear which model is instruction tuned v/s not.
>
> Understood, we will fix it.
>
> > Sec 6.2 presents an interesting experiment; but the figure needs to be improved. The explanations don’t match the legends of the figure (for instance, what is true/false). Also, do we mask the source sentence as well?
>
> We will improve the captioning of the figure. True/False refers to whether there are instructions provided (True) vs not provided (False). We do not mask any of the tokens, but remove the entire attention layer in Section 6.2 (as opposed to Section 3).

---

> > ### Comment · Reviewer_NuLE · 2024-08-13
> >
> > Thanks! My score remains the same. I do think that the motivations for task recognition should be explained better in the paper.

---

### Official Review · Reviewer_iAtv · 2024-07-20

**Soundness:** 3
**Presentation:** 2
**Contribution:** 2
**Rating:** 6
**Confidence:** 3

**Summary:**

This paper investigates where in-context learning occurs within large language models, focusing specifically on machine translation and code generation tasks. The authors introduce a "layer-from context-masking" technique to identify at which layer an LLM transitions from task recognition to task execution during ICL. They apply this method to several models including GPT-Neo2.7B, BLOOM3B, LLaMA7B (base and chat), and StarCoder2 (3B and 7B).

The key finding is that models do not need to maintain attention over the entire context throughout all layers to perform the task. Instead, there appears to be a "task recognition point" after which attention to the context is no longer necessary. This point varies between LLMs but generally occurs in the middle layers. The authors characterize this as a three-phase process: initial processing where masking has little effect, a critical phase where masking greatly impacts performance, and a final phase where additional context processing yields minimal gains.

The paper also explores the roles of instructions versus examples, the differences between instruction-tuned and non-instruction-tuned models, and whether these phenomena generalize across different tasks.
The authors highlight the potential practical implications of their findings, particularly for inference efficiency. They estimate that up to 45% computational savings could be achieved by eliminating unnecessary context processing in later layers.

**Strengths:**

- The authors introduce a "layer-from context-masking" technique to probe the internal workings of large language models during in-context learning tasks. This method offers a fresh perspective on how these models process and utilize contextual information, providing insights that were not previously available.
- The identification of a "task recognition point" and the characterization of a three-phase process in task recognition and execution are important.
- The finding that attention to context can be removed after certain layers without significant performance degradation has direct practical applications for improving inference efficiency in LLMs, potentially leading to substantial computational savings.

**Weaknesses:**

- One notable weakness is the limited exploration of why different models exhibit varying behaviors in terms of their "task recognition point" and critical layers. For instance, the authors observe that GPT-Neo has more severe critical layers compared to other models but do not provide a thorough analysis of potential reasons for this difference. A more in-depth investigation into the architectural differences, training data, or other factors that might contribute to these variations would significantly enhance the paper's insights and generalizability.
- The paper doesn't adequately address potential confounding factors that could influence the results. For example, the impact of different tokenization schemes across models or the potential effects of the specific prompt format used are not discussed in depth.
- The paper's focus on machine translation and code generation tasks, while valuable, raises questions about the broader applicability of the findings. The authors could strengthen their work by discussing potential limitations in generalizing these results to other types of tasks or by providing a theoretical framework that explains why these findings might (or might not) extend to other domains of language model application.
- The paper uses models of different sizes (2.7B to 7B parameters) but doesn't systematically analyze how model size affects the location of the "task recognition point" or the distribution of critical layers. A more structured comparison across model sizes could reveal important trends.

**Questions:**

1. You observe varying behaviors across different models, particularly in terms of the "task recognition point" and critical layers. Could you provide more insight into why these differences occur?
2. Your study focuses on machine translation and code generation. How confident are you that these findings would generalize to other types of NLP tasks?
3. Your study includes models of different sizes. Do you observe any consistent trends in how model size relates to the task recognition point or the distribution of critical layers?

**Limitations:**

In the NeurIPS Paper Checklist (after appendix), they mentioned that they have discussed the limitations in the conclusion section (Line#592). However, I didn't find any such thing.

---

> ### Author Rebuttal · Authors · 2024-08-07
>
> Thank you for the insightful comments on the paper and for the positive view of our work. We provide our response to the weaknesses (the questions are closely related)
>
> > One notable weakness is the limited exploration of why different models exhibit varying behaviors in terms of their "task recognition point" and critical layers. GPT-Neo has more severe critical layers compared to other models but do not provide a thorough analysis of potential reasons for this difference.
>
> Unfortunately, the differences are not due to easily observable hyperparameters like model size or architecture. To put in another way, why do large models exhibit different characteristics? Training data and training dynamics is a typical suspect. While very interesting, we feel that a thorough analysis is perhaps outside the scope of this paper.
>
> > The paper doesn't adequately address potential confounding factors that could influence the results. For example, the impact of different tokenization schemes across models or the potential effects of the specific prompt format used are not discussed in depth.
>
> We would like to discuss with the reviewer on the suggested confounding factors.
>
> * Tokenisation schemes, we’re not sure how an analysis on tokenisation scheme would be relevant to the problem being studied here. We used GPTNeo, Llama2, Bloom, Starcoder, all of which have different tokenisation schemes.
>
> * Specific prompt format. There are two main prompt formats used in the paper, reflecting the task.
>
> 1a. “Translate from {L1} to {L2}: Q: {source_sentence} A:,
> 1b. “Translate from {L1} to {L2}: Q: {source_sentence} A:,
>
>
> 2. Write a function to find the longest chain which can be formed from the given set of pairs. Q: {program_description}, A:
>
> We hope this gives the reviewer some confidence that there is sufficient coverage of the confounding factors highlighted.
>
> Instead, the potential confounding factors that we have investigated more pertinent to the research question are
> * With/without instructions
> * Number of In-context Examples
> * Different models Families
> * Instruction Tuned VS Not Instruction Tuned Models
> * 2 Different Tasks
>
> > The paper's focus on machine translation and code generation tasks, while valuable, raises questions about the broader applicability of the findings. The authors could strengthen their work by discussing potential limitations in generalizing these results to other types of tasks or by providing a theoretical framework that explains why these findings might (or might not) extend to other domains of language model application.
>
> Thank you for the suggestion, we will consider whether it’s possible to frame this better.
>
> > The paper uses models of different sizes (2.7B to 7B parameters) but doesn't systematically analyze how model size affects the location of the "task recognition point" or the distribution of critical layers. A more structured comparison across model sizes could reveal important trends.
>
> Thank you for the suggestion, we think this does not change the main point or contribution of the paper but acknowledge that it can be an interesting addition for extended experiments that we are happy to carry out
>
> > Limitations
>
> Most sincere apologies we seem to have missed this in the current draft... We would be extremely explicit about the scope of the experiments, and also include those the reviewer has highlighted. (exploration of different models, tasks, sizes).
>
> Currently in the paper, we have several "future work". i.e. limitations that we have not figured out.
> * _“It is not  immediately clear why GPTNEO has such critical layers and suffers compared to the other models, although we note that this is unlikely to be due to size or model architecture as BLOOM is also around the same size as GPTNEO and performs more similarly to LLAMA. We suspect that it could be due  to training data or some other factor related to the training dynamics but leave this for future work.
> “_ (line 245:249)
> * _"Potential reasons for this difference might be due to cross-entropy loss associated with task tuning for MT vs non-specific training on large corpora. We leave this as an open question for future work."_ (line 299-301)

---

### Decision · Program_Chairs · 2024-09-25

**Decision:**

Accept (poster)

**Comment:**

This paper provides a detailed study on where in the LLM architecture is a task-related information is learnt. Once the task is learned, the authors propose to mask the attention to in-context examples for that task for that layer onwards. This results in compute savings and increased interpretability. The paper is well written and the experiments are thorough. Based on revisions from previous submissions to other conferences the reviewers have improved the paper.